# *Helicobacter pylori* Dormant States Are Affected by Vitamin C

**DOI:** 10.3390/ijms24065776

**Published:** 2023-03-17

**Authors:** Paola Di Fermo, Silvia Di Lodovico, Emanuela Di Campli, Sara D’Arcangelo, Firas Diban, Simonetta D’Ercole, Mara Di Giulio, Luigina Cellini

**Affiliations:** 1Department of Pharmacy, University “G. d’Annunzio” Chieti-Pescara, 66100 Chieti, Italy; 2Department of Medical, Oral and Biotechnological Sciences, University “G. d’Annunzio” Chieti-Pescara, 66100 Chieti, Italy

**Keywords:** *Helicobacter pylori*, viable but non culturable, dormant state, vitamin C, persistent infections

## Abstract

*Helicobacter pylori* colonizes human gastric mucosa, overcoming stressful conditions and entering in a dormant state. This study evaluated: (i) *H. pylori*’s physiological changes from active to viable-but-non-culturable (VBNC) and persister (AP) states, establishing times/conditions; (ii) the ability of vitamin C to interfere with dormancy generation/resuscitation. A dormant state was induced in clinical MDR *H. pylori* 10A/13 by: nutrient starvation (for VBNC generation), incubating in an unenriched medium (Brucella broth) or saline solution (SS), and (for AP generation) treatment with 10xMIC amoxicillin (AMX). The samples were monitored after 24, 48, and 72 h, 8–14 days by OD_600_, CFUs/mL, Live/Dead staining, and an MTT viability test. Afterwards, vitamin C was added to the *H. pylori* suspension before/after the generation of dormant states, and monitoring took place at 24, 48, and 72 h. The VBNC state was generated after 8 days in SS, and the AP state in AMX for 48 h. Vitamin C reduced its entry into a VBNC state. In AP cells, Vitamin C delayed entry, decreasing viable coccal cells and increasing bacillary/U-shaped bacteria. Vitamin C increased resuscitation (60%) in the VBNC state and reduced the aggregates of the AP state. Vitamin C reduced the incidence of dormant states, promoting the resuscitation rate. Pretreatment with Vitamin C could favor the selection of microbial vegetative forms that are more susceptible to *H. pylori* therapeutical schemes.

## 1. Introduction

*Helicobacter pylori* has co-evolved with humans, becoming a successful pathogen that is able to colonize near 50% of the world’s human population [1]. *H. pylori* infection is generally asymptomatic but can involve the presence of gastric pathologies, such as chronic gastritis, peptic ulcers, and gastric malignancies, and their severity is related to the strain’s virulence and the host’s predisposition [1,2]. *H. pylori*’s capability to establish a long-lasting equilibrium with the human host underlines its skillfulness to adapt and its remarkable ability to evade and manipulate the host immune system [3]. Moreover, *H. pylori* shows a sophisticated capability to persist and establish chronic infections. In fact, *H. pylori* shows a wide and dynamic ability to colonize human gastric mucosa, producing biofilm and/or entering in a dormant state, strategies that permit its long-term self-preservation and enable it to overcome stressful conditions [4,5]. In particular, when entering a dormant state, *H. pylori* changes its cell morphology from a spiral to a coccoid shape, with a different cell wall structure that makes it difficult for the host immune system to recognize it (bacterial mimicry). In this microbial state, it produces reduced urease activity, low metabolism and intracellular ATP, and oxidative damage, maintaining its infectious capability and the expression of virulence factors that contribute to the development of chronic/persistent disease and exacerbating recalcitrance [5,6,7,8,9,10]. Studies have shown that *H. pylori* coccoids are 10 times more resistant to amoxicillin than spiral forms [11].

The dormant state in *H. pylori* is a survival strategy induced by a variety of adverse conditions, such as a temperature downshift, nutrient depletion, aerobic conditions, and the presence of antibiotics [12]; it allows the microorganism to survive in hostile environments, minimizing its detectability by conventional culture techniques [13,14]. Studies show different states of dormancy: the viable but non-culturable (VBNC) state and antibiotic persisters (AP) [15]. These two states display some similarities and differences and have long been the subject of debate. The VBNC state represents a microbial mechanism of survival against external stresses such as starvation, intolerable temperatures, antibiotics, sub-optimal oxygen levels, salinity, and pH levels [16,17,18]. The AP state describes a cellular sub-component that, in the presence of lethal doses of antibiotics, is able to survive antibiotic treatment and represents less than 1% of original population. The main difference between the two dormant states is related to their resuscitation dynamic: VBNC microbes are not capable to regrow on nutrient media after the removal of stress, probably because they need longer resuscitation times, while AP microbes are able to regrow on nutrient media after the removal of the antibiotic stress after a lag period. As proposed by Ayrapetyan et al., VBNC and AP may be physiologically related and considered as part of a “dormancy continuum” [19,20].

These broad stress responses underline the ability of these microbes to evade the action of antimicrobial therapies, strongly suggesting the need to identify new strategies that target dormant cells in order to improve the eradication rate.

Vitamin C (VitC) is a powerful antioxidant, an essential micronutrient for human health, an immune enhancer as a protective agent or alternative to therapy, an urease inhibitor, and a collagen- and prostaglandin-stimulating agent [21,22,23]. Low concentrations of VitC are associated with bleeding, delayed wound healing, anemia, cancers, and microbial infections [24,25,26]. In particular, gastritis and bleeding from gastric and duodenal ulcers are related to ascorbic acid deficiency, and low levels of VitC in gastric juice and serum are linked to patients infected by *H. pylori* [27,28,29]. Many studies have shown that, in combination with antibiotic treatment, VitC can increase the efficiency and eradication of *H. pylori* in infected patients (200 mg/day–500 mg/day or BID) [30,31,32], although these data are controversial [33].

Furthermore, a recent study suggests that VitC can reactivate dormant forms of *Mycobacterium tuberculosis* by stimulating the cellular respiration process [34]. The new population in its vegetative form is easier to treat, producing a favorable eradication rate. 

Based on all these premises, the aim of this work was to understand in greater depth the dormant state of *H. pylori*, evaluating the potential role of VitC in microbial dormancy. This investigation of how *H. pylori* enters dynamic dormant states, and the evaluation of VitC interference in these physiological changes, can offer novel solutions for more effective therapies. In the first step, we evaluated the progression of physiological changes from the metabolic active state (related to a spiral shape) to dormant forms (related to a coccoid shape), observing, establishing, and differentiating the correct times and conditions to induce dormant phenotypes in *H. pylori.* Subsequently, we investigated the role of VitC on the two dormant states of *H. pylori*, evaluating its capability to interfere with the generation and resuscitation of dormant states; the following flowchart displays the study design (Figure 1).

## 2. Results

In the first part of our work, we evaluated the progression of physiological changes in *H. pylori* from an active state to two dormant states (viable but non-culturable (VBNC) and antibiotic persister (AP)), establishing the times and conditions necessary to enter into these states. 

To do this, three conditions were evaluated: (1) the VBNC protocol: to generate VBNC cells, nutrient starvation was induced by incubation in two different minimal media, (1i) an unenriched Brucella broth (BB) and (1ii) a saline solution (SS). (2) The AP protocol: to induce persister cells; (2i) treatment with lethal amoxicillin (10 × MIC AMX). (3) Control: the control experiment was conducted in BB + 2% fetal calf serum. All conditions were monitored after 24 h, 48 h, and 72 h, at 8 days and 14 days, in terms of: Log CFU/mL determination, the total number of *H. pylori* cells (OD_600_), an MTT assay metabolic test for viability quantification, and microscopic observations for morphology/viability detection with Live/Dead (Syto9, PI) staining.

As reported in Figure 2A, (Log CFU/mL), in the control, a progressive increase in CFU/mL of *H. pylori* was detected from t = 0 to 72 h (from 6.43 to 7.2 Log CFU/mL), followed by a reduction in CFU/mL after 8 days and 14 days; this describes the normal trend in *H. pylori* growth (4.5 Log CFU/mL, 1.5 Log CFU/mL, respectively). 

Following the VBNC protocol, *H. pylori* was incubated in a minimal medium (BB conditions). The growth trend was different in respect to the control, resulting in a slow reduction in CFU/mL of *H. pylori* with a maximum value of 5.4 Log CFU/mL after 72 h, followed by a greater growth decrease after 8 and 14 days. In the saline solution (SS conditions), a rapid CFU/mL reduction was recorded, in respect to the control, of 3 Log units both at 24 h and 48 h, reaching 2 Log after 72 h of incubation and 0 CFU/mL after 8 and 14 days.

Following the AP protocol (the broth culture), *H. pylori* was incubated in the presence of 10 × MIC of amoxicillin (AMX condition); after 24 h, its concentration was reduced by 4 Log units (99,99% reduction), decreasing to 0 CFU/mL after 48 h of incubation.

A similar trend was observed in Figure 2B (OD_600_), which shows the total number of *H. pylori* cells cultured under different conditions. The highest OD_600_ values were recorded in the control and in the BB conditions. Lower values were recorded for SS and AMX conditions, with a weak increase in OD_600_ values after 48 h for both the SS and AMX conditions, followed by a rapid reduction after 72 h. In all studied conditions, after 72 h of incubation, a general period of stasis was detected in which no decrease or increase in absorbance values was recorded.

Regarding the MTT assay (Figure 2C), the metabolic activity of bacterial cells in the control increased, reaching its highest value after 48 h of incubation, which was related to the highest value of CFU/mL. In the BB condition, no significant modification of metabolic activity was recorded, following the same trend as the CFU/mL amount during the incubation period. In the SS and AMX conditions, the MTT values decreased with a percentage of reduction ranging from 23% (24 h) to 55% (14 days) and from 74% (24 h) to 93% (14 days), respectively.

The analysis of *H. pylori* morphotypes and their viability is shown in Figure 3. In the control, during the incubation time, the morphotype of the *H. pylori* population remained the same: spiral-shaped forms that were Syto-9 positive and PI negative, suggesting that bacteria were viable with a small number (10%) of coccoid forms with red fluorescence (dead cells). After 14 days, the bacillary forms remained the predominant morphotype (90%) but were PI positive; in fact, we recorded 60% spiral-shaped dead cells and 30% spiral-shaped viable cells with 10% coccoid dead cells. In the BB condition, at all control times, the coccoid morphotype with viable spiral shaped forms was predominant until eight days, with a parallel decrease in the total population.

In the saline solution (SS condition), during the experimental period, a prevalent coccoid morphotype was observed with a continuous increase in coccal cells, which reached its highest percentage (98%) after 8 days, with 96% coccal viable cells (Syto9 positive) and 2% coccal dead cells (PI positive). A stronger reduction (48%) in viability was observed after 14 days.

All detected data were statistically significant (*p* < 0.05) with respect to the control.

Upon treatment with a lethal concentration of AMX at 48 h, the remaining bacterial population was the coccoid morphotype, organized in large aggregates of *H. pylori* viable coccal cells. After 72 h, no significant modification was observed in terms of bacterial morphotype and viability.

The data provide the following times and conditions for *H. pylori* to transition from a vegetative state to a dormant state. In the BB state, (1i) bacteria did not lose cultivability over time, so the conditions of growth in this minimal medium were excluded from our experimental design. In the SS condition, (1ii) nutrient starvation was induced. After eight days, the bacteria’s ability to grow on media was lessened, exhibiting lower, although still present, metabolic activity; this represents the best conditions and time to generate the VBNC state. For the AP cell generation, based on these results, 48 h was defined as the necessary time after lethal AMX treatment (2i), in which bacteria lost the capability to produce CFUs in media but remained viable for some time; this represents the best time and conditions for the generation of AP bacteria.

In the second part of this work, we investigated VitC’s ability to: (i) interfere with dormancy generation (VBNC and AP) and (ii) resuscitate dormant VBNC and AP *H. pylori* cells.

First, the antibacterial effect of VitC was determined against the *H. pylori* strain 10A/13. More specifically, the MIC and MBC concentrations were the same, 2048 mg/L, and, in the experiments, VitC was used at ¼ MIC (512 mg/L). When included as a control, VitC did not significantly affect *H. pylori* growth.

In order to evaluate VitC’s ability to interfere in VBNC and AP generation, VitC was added before (t = 0) the generation of the two dormant states.

For the VBNC state generation, VitC seems to affect *H. pylori*’s adaptive morphological transformation into a coccoid form, reducing its capability to respond to nutritional stress; the bacteria remaining in the spiral-shaped form seem to be more vulnerable to nutrient starvation. In fact, with respect to the VBNC cells generated after 8 days in SS (control VBNC, 95% viable coccoid form and 5% dead coccoid form), bacteria incubated in SS + VitC for 8 days were not able to produce CFU/mL, remaining viable, as shown by the Live/Dead and staining MTT assay (Figure 4A), but they were less able to convert to the coccoid morphotype (see arrows). In particular, 50% of coccal forms were recorded as being mainly viable (90%) and 50% of spiral forms were dead (90%). The same results were also observed after 9 days, 10 days, and 11 days in both conditions (control and treated with VitC). 

To evaluate its potential role in the resuscitation process, VitC was added to the VBNC cells after eight days of incubation. The VBNC cells in the presence of VitC were monitored after 24 h (9 days), 48 h (10 days), and 72 h (11 days). As reported in Figure 5, in the presence of VitC, there was a regrowth (2 × 10^3^ CFU/mL ± 4 × 10^2^) of VBNC cells after 24 h and 48 h. Additionally, after 72 h of treatment of VBNC with VitC, a regrowth of the culturable cells was observed, with a parallel increase in MTT values (OD values, from 0.19 ± 0.03, 8 days to 1.5 ± 0.1, 9 days; 2.3 ± 0.5, 10 days; 1.8 ± 0.5, 11 days). Particularly evident was the increase in the amount of viable bacillary forms of *H. pylori* compared to the coccoid viable cells when VitC was added to the VBNC *H. pylori* population (Figure 5, arrows). For both *H. pylori* phenotypes, 20% of the population was in spiral form (60% viable and 40% dead) and the other 80% was coccoid (90% viable and 10% dead). During the observation period (11 days), an increase in the bacillary population was detected, leading to the hypothesis that VitC may have a significant effect on the morphological transition of the coccoid dormant form into a spiral/bacillary active form. The VBNC state is also associated with bacterial aggregates. Additionally, it was observed that, in the presence of VitC, there are few aggregates or clusters (Figure 5).

For the AP state generation, VitC seems to inhibit the transition of *H. pylori* into the antibiotic persister state. In fact, at 48 h of incubation with AMX and VitC, not all of the bacterial population had entered the dormant phase: 1% of population remained in a spiral bacillary form and were viable and culturable (3.7 × 10^4^ CFU/mL ± 5 × 10^3^). The viable coccoid cells remained the most common bacterial phenotype after a further 24 h (72 h) and 48 h (96 h) of incubation (Figure 6). In this dormant state (AMX + VitC for 48 h), we again attempted treatment with 10 × MIC AMX to induce the death of the 1% of the bacillary population and further stress the bacterial population; during this time, the whole bacterial population was PI positive, both in bacillary and coccoid forms, inducing amorphic forms and the production of cellular debris. The second cycle of AMX treatment guaranteed the complete killing of the bacterial population.

To study the effect of VitC on the AP state, the VitC was added to the AP population and incubated for 24 h and 48 h. VitC was not able to resuscitate the AP cells but was capable of causing the clustering of persister cells, reducing their number. In fact, from microscopic observations, it was found that AMX induced the production of viable coccoid cellular aggregation, and VitC seems to have a disaggregation effect on the bacterial population. The morphotype was 100% coccoid and the population morphotype remained the same as the control sample (AP cells) (Figure 7).

## 3. Discussion

*Helicobacter pylori* infection is one of the most common infections worldwide: this bacterium is capable of infecting approximately 50% of the world’s population, and up to 70% in developing countries [35]. In the absence of antibiotic therapy, it generally persists in the human stomach and duodenum. It is usually acquired during childhood and establishes lifelong chronic progressive gastric inflammation, leading to clinical complications in less than 10% of infected individuals. The infection has been linked to gastric and duodenal ulcers in 1–10% of infected patients, gastric carcinoma (0.1–3%), and gastric-mucosa-associated lymphoid tissue (MALT) lymphoma (less than 0.01%) [36].

*Helicobacter pylori* is mainly present in a classical spiral-shaped vegetative form when detected in human gastric biopsy specimens. Meanwhile, it has evolved special adaptive mechanisms to survive extreme adverse situations both outside the host and in the human organism, acquiring the ability to lose its typical spiral-shaped form and convert into latent dormant coccoid morphotype [37]. In this study, was studied the manifold *H. pylori* 10A/13 behaviors that counteract stressful conditions.

Specifically, we showed that the coccoid form is indeed a manifestation of cellular adaptation to enable survival in suboptimal conditions, such as nutrient starvation (saline solution) induced after eight days, with the cellular conversion of the bacillary forms into viable coccoid forms, which were unable to grow on media (VBNC) and maintained low metabolic activity. Additionally, Aktas et al. showed that *H. pylori* bacillary cells, incubated in PBS (nutrient starvation) at 37 °C, were completely transformed into the viable coccoid morphotype after eight days [11]. For the generation of coccoid persister cells, we used amoxicillin (AMX), one of the antibiotics typically used for the treatment of *H. pylori* infections [38]. Our results showed that AMX can induce the coccoid morphotype of *H. pylori* 10A/13. These data are in agreement with Berry et al., who showed a marked increase in the number of coccoid cells of *H. pylori* after treatment with 10xMIC of AMX [39]. However, the authors did not discriminate between the living and dead coccoid populations. In this study, it was observed that, after 10xMIC of AMX treatment for 48 h, a morphological conversion was induced in coccoid cells that remained viable. As reported by Faghry et al., three antibiotics (metronidazole, clarithromycin, and amoxicillin) were able to induce *H. pylori* coccoid forms, but the highest rate of morphological modification was induced by AMX treatment after 72 h. They supposed that, by acting on the cell wall penicillin-binding protein, AMX could have a role like a stress factor, affecting morphological modification by the induction of enzyme peptidase, which promotes the cell wall’s modification into a coccoid form [38]. In addition, as described by Costa et al., the transition to coccoid forms is related to the accumulation of the N-acetyl-D-glucosaminyl-*β*(1,4)-N-acetylmuramyl-L-Ala-D-Glu motif in *H. pylori*. Therefore, this can play a role in the bacterial resistance to AMX. Moreover, Costa et al. demonstrated that, during the morphological conversion of *H. pylori*, similarly to the sporulation of *Bacillus sphaericus,* glutamyl-diaminopimelate endopeptidase was activated; this phenomenon is related to the genesis of resistant coccoid forms. Therefore, the high resistance of coccoid forms could be related to the different compositions of cell wall in coccoids and the subsequent loss of antibiotic targets in these forms [40]. In general, our data underline that the fully formed coccoid cells (AP and VBNC cells) are viable forms. In the VBNC and AP protocols, during the generation of dormant states, we observed a morphological change of *H. pylori* from the bacillary to the coccoid forms during the experimental period (8 days for VBNC and 48 h for AP). The significant decrease in the number of CFUs was not mirrored by the loss in bacterial viability, as shown by the Live/Dead staining assay; this is likely due to the fact that *H. pylori* 10A/13 coccal forms are viable but unable to grow on media, with a low metabolic activity and preserved membrane integrity.

In most cases, the administration of antibacterial drugs leads to the complete eradication of *H. pylori*; however, the presence of resistant bacterial strains, as well as tolerant *H. pylori* forms (biofilm and dormant forms), can interfere with their complete eradication. In particular, the tolerant dormant state is difficult to treat with classical therapeutic schemes; in fact, the ultrastructural changes in the cell membrane and in the metabolic pathways reduce drug efficacy [41].

To improve the *H. pylori* eradication rate, plant extract, bovine lactoferrin, probiotics, and antioxidants have been investigated as potential adjuvants in innovative therapies for the management of *H. pylori* infection [42,43].

The colonization of gastric mucosa by *H. pylori* leads to inflammatory processes, producing a significant amount of reactive oxygen intermediates and causing the activation of immune cells, leading to mucosal destruction [44,45]. Chronic *H. pylori* infection can lead to peptic ulceration and may be a potential risk factor for gastric carcinoma. As confirmed by Buommino et al., the host cell response against chronic oxidative stress is hampered by *H. pylori*-infected cells, due to the increased activity of antioxidant enzymes during *H. pylori* infection [46]. El Mortaji et al. discovered that a peptide of the type I Toxin−Antitoxin (TA) system induces *H. pylori*’s morphological change from a spiral shape to a coccoid shape, suggesting that oxidative stress is responsible for the coccoid formation due to its repression of the antitoxin promoter, leading to the increased production of AapA1 toxin expression. These findings reveal that the TA system plays a role in dormant/persister *H. pylori*, enhancing its tolerance to antibiotics [47].

Several clinical studies have shown that lower VitC levels, both in gastric juice and serum, are linked to patients with *H. pylori* gastritis and peptic ulcers [28,29]. Normally, VitC is actively secreted from plasma to gastric juice. As suggested by Hussain et al., VitC is a preventative and therapeutic agent against *H. pylori* due to its ability to inactivate the urease enzyme, a potent virulence factor that is crucial for the bacterium’s survival in an acidic gastric environment [30]. High concentrations of VitC in gastric juice favor the reduction in Ni++ centers coordinated to the histidine residues of the urease, which are secreted from *H. pylori*, leading to the inactivation and denaturation of the enzyme and preventing *H. pylori* from surviving and colonizing the acidic stomach [48,49]. Moreover, VitC represents a cofactor for the synthesis of type IV collagen, which is essential for lamina propria synthesis in the stomach. Stronger collagen could cause difficulties for the infiltration of the gastric bacteria [30].

For this reason, antioxidants, such as VitC, are thought to be possible effective agents that could be administered in conventional treatments.

On the basis of these assumptions, in the second part of this study, we demonstrated the ability of VitC to interfere with dormancy generation (both VBNC and AP generation) and to resuscitate the dormant *H. pylori* 10A/13.

The results obtained in this study demonstrated for the first time VitC’s ability to reactivate the VBNC coccoid *H. pylori* 10A/13 cells, inducing the morphological transition from coccoid to bacillary shapes in culturable cells. The increase in the cellular respiration process was proposed as a possible VitC stimulus in a recent study by Vilchèze et al. [34]. Moreover, cells forming aggregates were observed during the dormancy period [50]. We have shown that the aggregation capacity decreases in the presence of VitC. This phenomenon was particularly prevalent against the antibiotic persister population, which, after treatment with VitC, maintains its vitality (only 1% of the population revert to a spiral viable form) but loses its ability to form clusters. The difficulty in eradicating these persister cells leads to the establishment of a latent *H. pylori* infection and drug-resistant mutants. However, in our study, the disaggregated persister population seemed to be more vulnerable to the biocidal action of a second cycle of amoxicillin, which leads, after 24 h, to the death of both the bacillary and coccoid forms, inducing amorphic forms and the production of cellular debris.

The chronic infection of *H. pylori* is related to its capability to resist antibiotics due to the high incidence of MDR *H. pylori* strains. This bacterium is also capable, by a chameleon-like behavior, to express a tolerant phenotype such as biofilm and dormant states. It is well known that oxidative stress plays a crucial role in the tolerant *H. pylori* phenotype. Reactive oxygen species (ROS) secreted by the host immune cells against *H. pylori* infection cannot effectively eliminate the pathogen [51]. Moreover, ROS can promote the *H. pylori* biofilm formation as well as its morphological transformation by toxin-antitoxin type I system activaction [47]. It is therefore possible to attribute to VitC a potential role in preventing entry into tolerant forms and in the resuscitation of dormant forms (VBNC and AP).

As reported by Reshetnyak et al. [5], some bacterial cytokines, including the 16–17 kDa protein named “resuscitation-promoting factors” isolated from bacteria such as *Micrococcus luteus*, promote the “revival” of resting cells, acting like a pheromone belonging to the bacterial cytokines. It is probable that VitC plays a role in the production of bacterial cytokines or in activating some heat shock proteins capable of initiating the reversion of dormant forms in *H. pylori*.

The results reported here should be considered in the light of a limitation related to the use of one *H. pylori* strain for the experiments, although the chosen clinical strain was selected both for its multi-drug-resistant (MDR) profile and its virulence factors. The bacterium was isolated in a patient with active chronic gastritis who had never before been treated for *H. pylori* infection. These characteristics are related to a strain that is capable of establishing the best strategies for interacting with the human host.

Further studies are needed to confirm the effects of several strains of *H. pylori* and to better understand VitC’s mechanisms of action on its dormant state.

## 4. Materials and Methods

### 4.1. H. pylori Strain and Growth Conditions

An *H. pylori* 10A/13 clinical isolate from the private collection of the Bacteriological Laboratory of the Pharmacy Department, University “G. d’Annunzio” Chieti-Pescara, was used in this study (ID Number RICH9RTLH). The bacterium was isolated from a patient with active chronic gastritis who had never previously been treated for an infection associated with *H. pylori.* The strain, previously characterized for its antimicrobial susceptibility profiles against the antibiotic commonly used in therapy [42], was an MDR strain, displaying a resistance profile against at least three antimicrobial classes (the strain is *cag*A positive, *ice*A1 negative, and *vac*A s1m2i1). The bacterium was cultured on a Columbia agar base (Oxoid, Milan, Italy) with 10% (*v*/*v*) laked horse blood plus IsoVitalex 1% (*v*/*v*) (BBL, Microbiology System, Milan, Italy); it was then incubated for 72 h at 37 °C in microaerobic conditions (5% O_2_, 10% CO_2_, 85% N_2_).

For the experiments, pure *H. pylori* 10A/13 colonies taken from a solid medium were inoculated in Brucella Broth (BB) containing 2% fetal calf serum (FCS) (Biolife Italiana, Milan, Italy) and adjusted to an optical density at 600 nm (OD_600_) of 0.2, corresponding to ~1.8 × 10^6^ CFU/mL for the experiments. Before the experiments, the regular helical/bacillary morphology, as well as the viability, were ascertained by Live/Dead staining (Molecular Probes, Invitrogen detection technologies, Eugene, OR, USA) under fluorescent microscopy (Leica Microsystems, Milan, Italy).

### 4.2. Evaluation of Vitamin C and the Amoxicillin Susceptibility Test

Vitamin C (VitC) and amoxicillin (AMX) were purchased from Sigma Aldrich Company (Sigma-Aldrich, St. Louis, MI, USA) and resuspended in BB plus 2% FCS (Biolife Italiana, Milan, Italy) to obtain final concentrations of 164,000 μg/mL and 2 μg/mL, respectively. The minimum inhibitory concentration (MIC) value of VitC against the standardized broth culture of *H. pylori* was determined using a microdilution method assay in 96-well microtiter plates (Nunc, Euro Clone SpA, Life Sciences-Division, Milan, Italy) [52]. Then, 2-fold dilutions of VitC and AMX stock solutions, ranging from 8200 to 64 µg/mL, and from 1 to 0.015 μg/mL, respectively, were prepared in BB plus 2% FCS. Then, 100 µL of VitC or AMX at each concentration and 100 µL of standardized bacteria were dispensed in each well of the 96-well microtiter plate and incubated in microaerobic conditions for 72 h at 37 °C. MIC values were measured by determining the lowest concentration of Vit C able to inhibit the visible growth of the microorganisms. The minimum bactericidal concentration (MBC) was determined by the sub-cultivation of 10 µL of suspensions from the non-turbid wells on chocolate agar (CA) and incubated as describe above. The MBC represents the lowest concentration of VitC that inhibited bacterial growth on the plates. The MBC values were also confirmed by an iodo-nitro tetrazolium violet assay (INT, Sigma-Aldrich, St. Louis, MI, USA), following the addition of 0.2 mg/mL of INT and incubation at 37 °C for 2 h. Viable bacteria reduce the yellow dye to a pink-purple, and dead cells do not produce a color change [52].

### 4.3. Induction of Dormancy in H. pylori with Different Methods

For the experiments, as described above, pure *H. pylori* 10A/13 colonies taken from a solid CA medium were inoculated in 50 mL BB containing 2% FCS and adjusted to an optical density at 600 nm (OD_600_) of 0.2 corresponding to ~1.8 × 10^6^ CFU/mL. The dormant state was induced following two methodologies: nutrient starvation (VBNC state) and antibiotic treatment at high concentrations of 10 × MIC AMX(AP state) [14].

For the *H. pylori* viable-but-non-culturable (VBNC) state protocol, we tested two methods of nutrient starvation, incubating it in: (i) an unenriched growth medium (BB without 2% FCS); (ii) a sterile saline solution (SS-0.9% NaCl). *H. pylori* 10A/13 cultivated in BB plus 2% FCS was used as the control.

For the *H. pylori* antibiotic persister (AP) state protocol, 10 × MIC AMX was added to the *H. pylori* 10A/13 culture. *H. pylori* 10A/13 cultivated in BB plus 2% FCS was used as the control.

All *H. pylori* cultures in each growth condition were incubated at 37 °C in microaerobic conditions for 14 days.

To evaluate the best timescale for the generation of dormant states, the samples taken from each condition tube were monitored after 24 h, 48 h, 72 h, 8 days, and 14 days by optical density (OD_600_), CFUs determination, Live/Dead staining and observation under fluorescent microscopy, and an MTT (Sigma-Aldrich, St. Louis, MI, USA) viability test, for the evaluation of the total count, bacterial cultivability, coccoid/bacillary morphology, and cell viability.

### 4.4. Determination of Vitamin C’s Ability to Interfere with H. pylori Dormancy Generation

In order to determine VitC’s ability to interfere with the generation of VBNC and AP, ¼ MIC of VitC was added to the *H. pylori* suspension before the generation of the two dormant conditions (t = 0). The detected best times for VBNC and AP generation were 8 days and 48 h, respectively.

Specifically, for VBNC generation, ¼ MIC Vit C was added to the *H. pylori* suspension in the saline solution and incubated at 37 °C in microaerobic conditions for 8 days (the best time for VBNC generation); for the AP generation, ¼ MIC Vit C was added to the *H. pylori* suspension in BB +2% FCS + 10xMIC AMX and incubated at 37 °C in microaerobic conditions for 48 h (the best time for VBNC generation).

After the incubations, the optical density (OD_600_), CFUs determination, Live/Dead staining and observation under fluorescent microscopy, and MTT viability test were conducted for comparison with the controls: untreated samples (BB + 2% FCS), untreated samples plus 1/4 MIC VitC (BB + 2%FCS + VitC), the SS-exposed bacteria without VitC (VBNC control), and the AMX-exposed bacteria without VitC (AP control).

### 4.5. Determination of Vitamin C’s Ability to Resuscitate H. pylori Dormant States

In order to determine the resuscitation effect of VitC on *H. pylori* dormancy, the VBNC and AP cells, generated as described above, were treated with ¼ MIC VitC. In detail, ¼ MIC Vit C was added to the VBNC bacterial suspension and grown in SS for 8 days; likewise, it was added to the AP bacterial suspension and grown for 48 h in BB + 2% FCS + AMX. All bacterial suspensions were incubated at 37 °C in microaerobic conditions and monitored at 24 h intervals for 72 h.

At each time control, the optical density (OD_600_), CFUs determination, Live/Dead staining and observation under fluorescent microscopy, and MTT viability test were evaluated in comparison to the controls: untreated samples (BB + 2% FCS), the SS-exposed bacteria without VitC (VBNC control), and the AMX-exposed bacteria without VitC (AP control).

### 4.6. H. pylori Total Bacterial Counts, Cultivability and Viability Determination, and Microscopic Analysis

In order to determine the total count of *H. pylori* under different conditions, samples taken from the bacterial culture were measured in a UV spectrophotometer at 600 nm. The different samples were also serially diluted in sterile PBS and the *H. pylori* cells were quantified by CFUs determination on CA and incubated in microaerobic conditions at 37 °C. The CFU/mL was counted after 3–5 days.

The viability of cells exposed to various conditions was assessed through Live/Dead staining (Molecular Probes, Invitrogen detection technologies, Eugene, OR, USA) as indicated by the manufacturer. The images were observed using Leica 4000 DM fluorescent microscopy (Leica Microsystems, Milan, Italy). For this purpose, aliquots of 500 μL were taken from each culture and centrifuged at 5000 rpm for 10 min to obtain the pellets. Afterwards, the pellets were stained with the Live/Dead staining kit, then mixed for 15 min in the dark, and examined under the fluorescence microscope. The morphology of cells was also examined under an inverted-phase contrast microscope (Leica 4000 DM). For the evaluation of the microbial shape, ten different fields of three slides were counted by three microbiologists (P.D.F., S.D.L. and M.D.G.).

The metabolic activity assay was performed on 96-microwell plates, using a 3-(4,5-dimethylthiazol-2-yl)-2,5-diphenyltetrazolium bromide (MTT) test. After each treatment, bacterial aliquots were incubated with 20 μL of MTT (5 mg/mL) for 4 h every 24 h. After centrifugation at 8000 rpm for 10 min, the pellet was resuspended in dimethyl sulfoxide for the extraction of the formed formazan dye, and then the absorbance was recorded using a microplate (ELISA) reader.

### 4.7. Statistical Analysis

Data were obtained from at least three independent experiments performed in triplicate. Data are shown as the means ± standard deviation (SD). Differences between groups were assessed with the paired Student’s *t*-test. *p* values ≤ 0.05 were considered statistically significant.

## 5. Conclusions

Our findings reveal that VitC both reduces *H. pylori*’s entry into dormant states and enhances the resuscitation rate. In terms of applications, pre-treatment with VitC in *H. pylori* therapeutic schemes could favor the selection of microbial vegetative forms that are more susceptible to treatments, as well as preventing persistent infections, representing a significant intervention in human health.

## Figures and Tables

**Figure 1 ijms-24-05776-f001:**
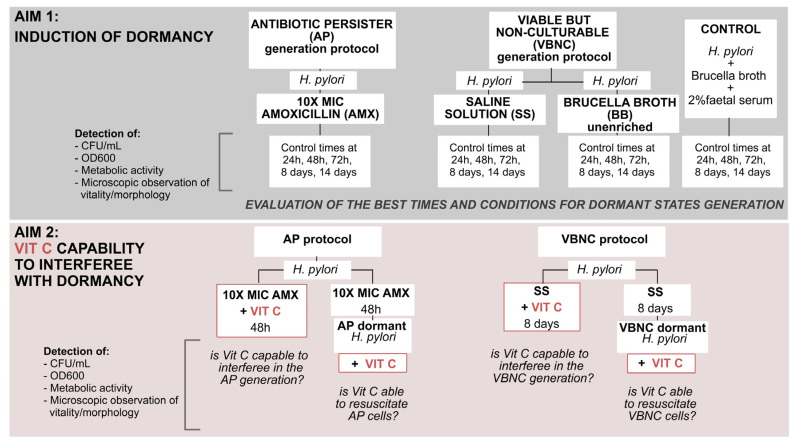
Study design and description of the aims of the study.

**Figure 2 ijms-24-05776-f002:**
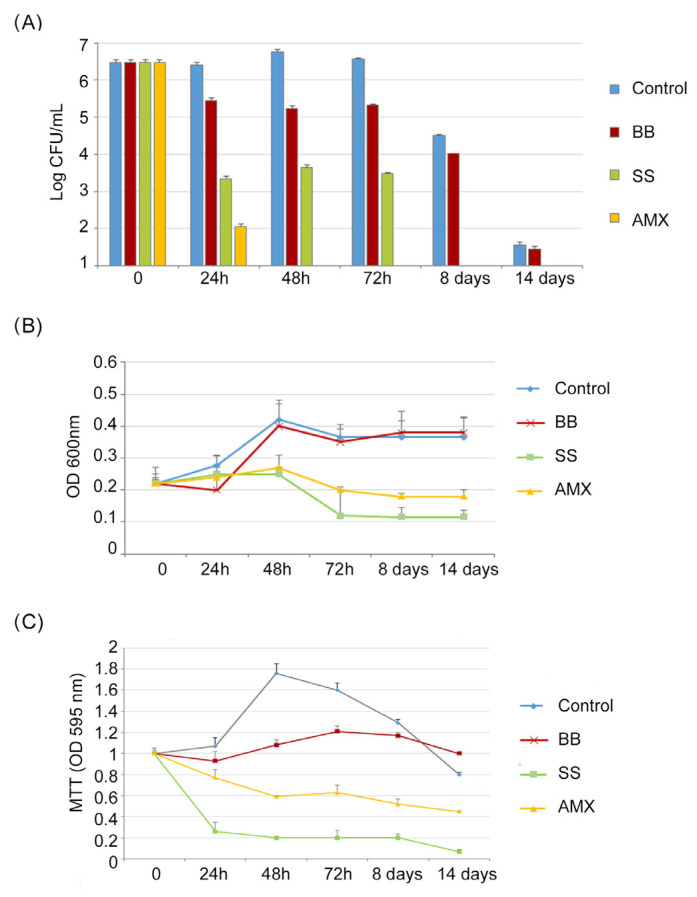
**Induction of dormancy**. (**A**) Culturable cell numbers of *H. pylori* incubated in a microaerobic atmosphere at 37 °C under different conditions. Control (BB + 2% FCS); BB (Brucella broth) and SS (saline solution) for the VBNC protocol; AMX (amoxicilllin 10xMIC) for the AP protocol recorded at t = 0, 24 h, 48 h, 72 h, 8 days, and 14 days. (**B**) Total number (OD_600_) of *H. pylori* incubated in a microaerobic atmosphere at 37 °C under different conditions, in time. (**C**) MTT assay of *H. pylori* for the evaluation of metabolic activity during incubation in a microaerobic atmosphere at 37 °C under the different conditions over time.

**Figure 3 ijms-24-05776-f003:**
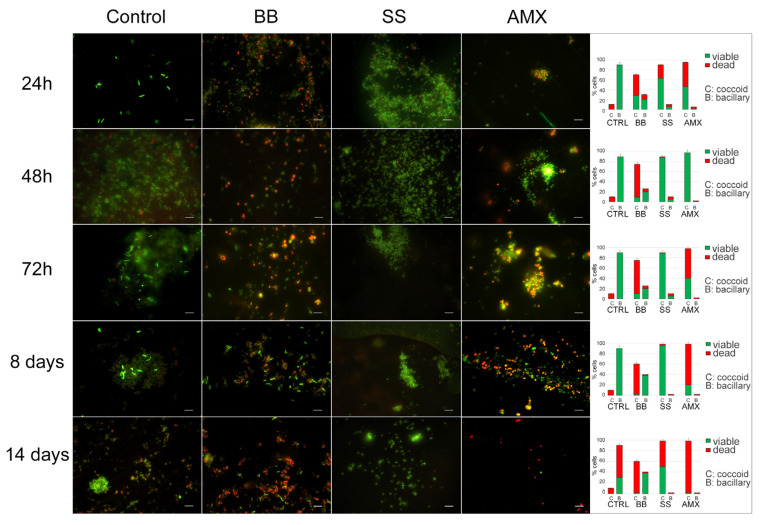
**Induction of dormancy.** Representative images of Live/Dead staining and the percentages of viable/dead spiral/coccoid morphotypes (graphics on the right) of *H. pylori* 10A/13 incubated in different conditions (BB, SS, and AMX) compared with the control, in time. The images observed by fluorescent Leica 4000 DM microscopy (Leica Microsystems, Milan, Italy) were recorded at an emission wavelength of 500 nm for SYTO 9 and of 635 nm for Propidium iodide, and several fields of view were randomly examined. Original magnification, 1000×.

**Figure 4 ijms-24-05776-f004:**
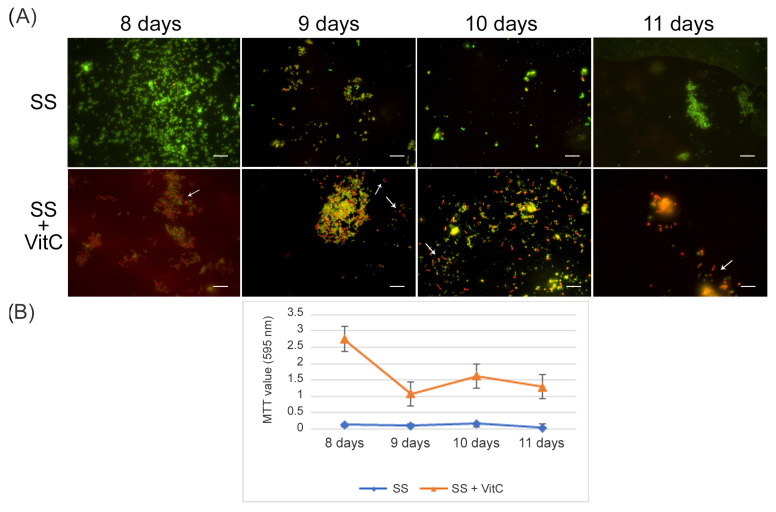
**VitC effect on VBNC generation**. The addition of VitC before (t = 0) the generation of *H. pylori* cells’ VBNC and monitoring after 8, 9, and 10 days of incubation. (**A**) Representative Live/Dead microscopy images of *H. pylori* 10A/13 incubated in SS in the presence of VitC for eight days, compared with the untreated sample (SS). Original magnification 1000×; (**B**) MTT values comparing the cell viability of VBNC cells generated in SS + VitC, with respect to the untreated sample (SS).

**Figure 5 ijms-24-05776-f005:**
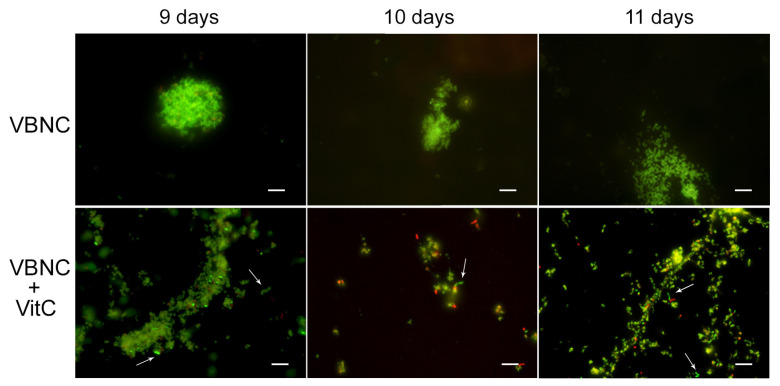
**Effect of VitC on the VBNC state.** Representative images of Live/Dead staining showing the effect of VitC on VBNC cells compared to the untreated samples (VBNC): VitC was added after 8 days of incubation after VBNC generation, and its effect was monitored for a further 24 h (9 days), 48 h (10 days), and 72 h (11 days); it was compared with the untreated sample (VBNC). The arrows indicate the viable bacillary forms in the presence of VitC. Original magnification, 1000×.

**Figure 6 ijms-24-05776-f006:**
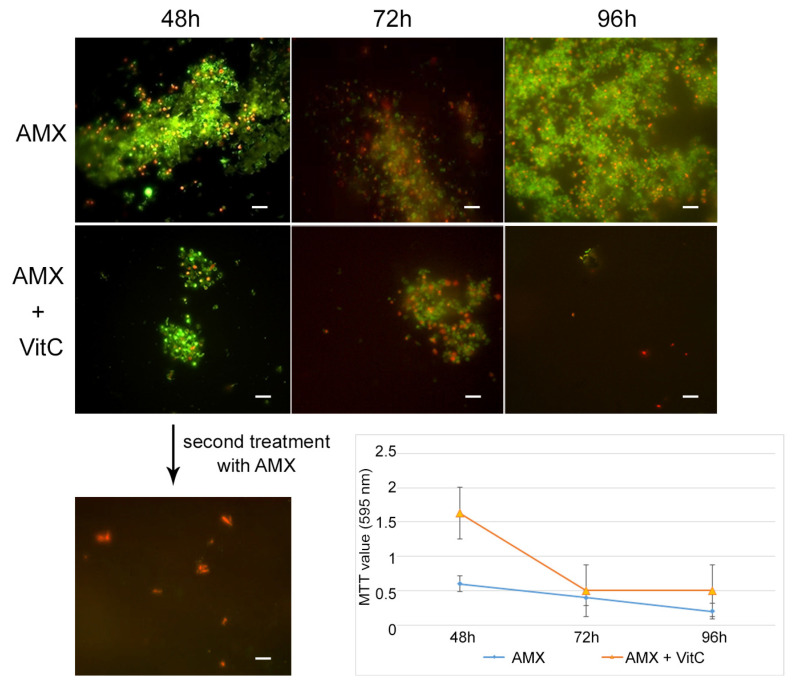
**VitC effect on AP generation**. Representative images of Live/Dead staining showing the effect of adding VitC to AMX on *H. pylori* cells during the AP generation. Monitoring occurred after 48 h, 72 h, and 96 h of incubation, and a comparison with the untreated sample was conducted (AMX). The second treatment with AMX guaranteed the complete killing of the population growing in AMX plus VitC for 48 h. Original magnification, 1000×. In the graphic, the MTT values show an increase in the viability of the *H. pylori* cells in AMX +VitC for 48 h, compared to those grown only in AMX (untreated sample).

**Figure 7 ijms-24-05776-f007:**
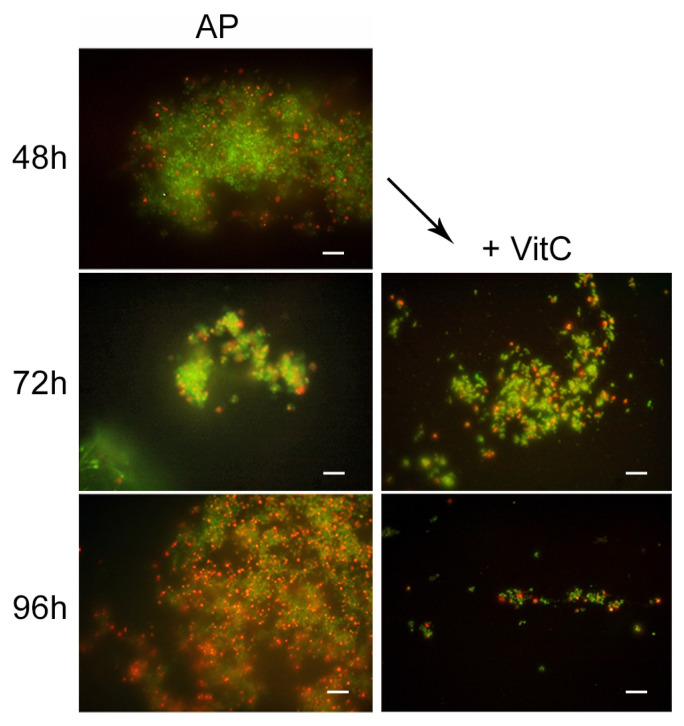
Representative images of Live/Dead staining showing the disaggregation action of VitC on AP cells, monitored after a further 24 h (72 h) and 48 h (96 h), compared to the untreated sample. Original magnification, 1000×.

## Data Availability

Not applicable.

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
