# Peer review of "Helicobacter pylori Dormant States Are Affected by Vitamin C"

_ijms, 2023, doi:10.3390/ijms24065776_

Round 1
Reviewer 1 Report
The original article by Paola Di Fermo et al. "Helicobacter pylori dormant states affected by vitamin C" is devoted to the study of the effect of vitamin C on the formation of a variety of forms of Helicobacter pylori (H.pylori) cultivated under various conditions in vitro. The impact of adverse factors leads to the fact that H.pylori undergoes structural and functional changes, loses its typical spiral shape and passes into a state of rest (dormancy). Active study and discussion in the scientific literature of dormant forms in non-spore-forming bacteria, which include most gram-negative microorganisms, including H.pylori, was noted at the end of the XX century (Kaprelyants AS, Gottschal JC, Kell DB. Dormancy in non-sporulating bacteria. FEMS Microbiol Rev. 1993;10(3-4):271-285). The authors consider coccoid forms as both Viable but Non-Cultured (VBNC) and dead states of bacteria. In the human body, VBNC forms of bacteria can reverse into a vegetative state. The authors convincingly demonstrated the effect of vitamin C on slowing down the transition of the H.pylori bacterium to a dormant form and a faster reversion to a vegetative form. Ultrastructural examination of coccoid forms of H.pylori revealed various cell wall defects that require repair before leaving the dormant state ("resuscitation", reversion) (Konstantinova, N.D., Zhukhovitskii, V.G., Didenko, L.V. et al. Ultrastructural Organization of Helicobacter pylori under Natural Conditions and during Ex Vivo Culturing. Bulletin of Experimental Biology and Medicine 131, 299–301 (2001). https://doi.org/10.1023/A:1017680205786). Vitamin C, being an integral part of hydroxylase enzymes, is actively involved in many metabolic processes. The authors suggest that vitamin C may also be involved in the restoration of the cell wall of H.pylori. Vitamin C also takes an active part in redox reactions, being a powerful antioxidant. In this connection, the authors suggest that it can contribute to reducing the amount of reactive oxygen species. And due to this, vitamin C improves the conditions for the vital activity of H.pylori.
Pheromones (bacterial cytokines such as Rpf) and some heat shock proteins can initiate the reversal of dormant forms of H.pylori. Perhaps vitamin C also plays a role in the production of bacterial cytokines.
The design of the study is well organized. The title, abstract and keywords correspond to the text of the article. Materials and methods are described in detail and clearly. Correct methods of statistical processing of the obtained data were used.
The data obtained by the authors are important for understanding the role of Helicobacter pylori for humans.
The authors refer appropriately to the most recent and up-to-date references.
Unfortunately, it is not possible to isolate separately VBNC and dead coccoid forms that are in the same culture. But there are also intermediate C- and U-shaped forms of the bacterium H.pylori. Don't the authors think that these intermediate forms are viable, but coccoid forms are non–viable (dead) forms?
Author Response
Reviewer 1
- We want to thank the Referee for the useful comments and for the work appreciation.
During the evolutionary course, H. pylori has refined surprising adaptive mechanisms, one of these is the ability to convert itself into coccoid forms which allows it to survive in inadequate conditions. Reshetnyak et al. (ref 5) suggest that there are intermediate morphological H. pylori forms: the C-shaped and/or U-shaped cells, passing from the spiral to the complete coccoid shape. These forms seem to coexist in the host gastric mucosa and can be also considered dormant forms. Our data underlines that the fully-formed coccoid cells (AP and VBNCcells) are viable forms. In the VBNC and AP protocols, during the generation of dormant states, we observed a morphological change of H. pylori from the bacillary to the coccoid forms during the time (8 days for VBNC, and 48h for AP). The significant decrease of the CFUs number was not mirrored by the loss of bacterial viability using the LIVE/DEAD staining assay, likely due to the fact that H. pylori 10A /13 coccal forms are viable but unable to grow on media, with a low metabolic activity, and preserved membrane integrity. In the new version of the manuscript, we emphasized this result inserting a new sentence (pag. 10, lines.293-300).
Reviewer 2 Report
Fermo et al reported Helicobacter pylori dormant state affected by vitamin C. The detailed methodology study demonstrated how to efficiently generate dormant state of H. pylori. The later part of this research showed vitamin C reduced the entry in VBNC state and AP state. However, the study only showed the phenotype screening result of vitamin C's effect on H. pylori. A detailed study/discussion of the mechanism of vitamin C's role need to be addressed to make a full and compelling story.
Besides, several suggestions may also help to strength the study:
1, all the studies were conducted on single strain 10A/13. The authors may need test more representative strains to show the effect is general.
2, Vitamin C's effect on normal cultured H. pylori is missing.
3, the morphology is important in determining the states of H. pylori . All the staining figures authors were showing are hard to tell spiral/coccid morphotypes.
4, Figure 4,5,6,7 are only showing representative images of Live/Dead staining. A detailed scientific quantification comparison need to be added.
Author Response
- We want to thank the Referee for the useful comments and we agree to the Referee suggestion.
- Toghether with the VitC mechanisms action explained in the introduction section (pag 2 lines 66-78) and in the discussion section (pag.9-10 lines 322-333 and 339-342.) in the new version of the MS, we inserted more comments related to the like pheromones VitC action (pag 11 lines 353-358).
- According to the Referee consideration, in each part of the discussion we specified that our data was obtained with H. pylori 10A /13. In discussion section, we inserted the following sentence relted to the potential limitation of the study (pag 11lines359-366). We also underlined the rational of the choice of this microrganism. Infact as indicated in the new version of the manuscript in M&M the chosen clinical strain was selected both for its MDR profile and virulence factors. Moreover, the bacterium was isolated in a patient with active chronic gastritis never treated before for an infection associated to H. pylori (the strain is cagA positive, iceA1 negative, vacA s1m2i1). All these characteristics are related to a strain capable to establish the best strategies to interact with the human host (pag 11, 371-376).
- The effect of VitC on normal pylori culture was not included in the MS. In the early step of our work, as control, we evaluated vitamin C’s effect on normal H. pylori culture. Vitamin C at ¼ MIC did not affect the normal H. pylori growth. The obtained data revealed that there was no significant reduction or increase of normal H. pylori cells growth in presence of vitamin C, compared to the control.
|
|
CFU/mL |
|
|
|
48h |
72h |
|
H. pylori control (BB+2% FCS) |
3,2*107 CFU/ml ± 1,3*107 |
6.6*107 CFU/ml ± 1.5*107 |
|
H. pylori in BB+2% FCS + Vitamin C (1/4 MIC) |
1,9*107 CFU/ml ± 0.09*107 |
5.2*107 CFU/ml ± 1.4*107 |
In the new version of the MS, in Materials and methods section, we inserted this control (pag 13, lines 437-438) and commented in the results (pag. 6 lines 178-179) the ineffective action
- According to the Reviewer comment, in the new version of the MS, we enlarged the figures to better show the representative images and we included new sentences in the results section: pag 6, lines 186, 204-205 and pag 7 lines 208-209. About the figure 4, we already have specified in the MS a quantification comparison between coccoid and spiral population, and the percentage of viable and dead in both morphotypes.
Reviewer 3 Report
Dear Editor,
thank you for giving me the opportunity to review this interesting paper.
The aim of this paper is to deepen the knowledge of the effect of VitaminC on Helicobacter pylori. The research team has demonstrated that VitaminC is able to avoid or delay the dormant state of H. pylori. This could have interesting implications on therapeutic approach against H. pylori infection.
The paper is well written and is easy to understand, the methodology is clear and adequate for the aim.
Minor revision:
- Can you maximize the quality of Figure 1? It is quite difficult to read it.
Author Response
- We want to thank the Referee for the useful suggestion and for the work appreciation.
We inserted a new higher quality Figure 1.
Round 2
Reviewer 2 Report
Thank you for addressing my questions and concerns.